# Development of a Probability-Based In Vitro Eye Irritation Screening Platform

**DOI:** 10.3390/bioengineering11040315

**Published:** 2024-03-26

**Authors:** Seep Arora, Anna Goralczyk, Sujana Andra, Soon Yew John Lim, Yi-Chin Toh

**Affiliations:** 1Department of Biomedical Engineering, National University of Singapore, Singapore 117583, Singapore; seepar17@gmail.com (S.A.); annaagoralczyk@gmail.com (A.G.); reddysujju710@gmail.com (S.A.); 2A*STAR Microscopy Platform, 61 Biopolis Drive, #06-20 Proteos, Singapore 138673, Singapore; john_lim@rsc.a-star.edu.sg; 3School of Mechanical, Medical and Process Engineering, Queensland University of Technology, Brisbane, QLD 4000, Australia; 4Centre for Biomedical Technologies, Queensland University of Technology, Kelvin Grove, QLD 4059, Australia; 5ARC Training Centre for Cell and Tissue Engineering Technologies, Kelvin Grove, QLD 4059, Australia

**Keywords:** eye irritation test, cell micropatterning, TRPV1 receptor, nociception, 3Rs tool

## Abstract

Traditional eye irritation assessments, which rely on animal models or ex vivo tissues, face limitations due to ethical concerns, costs, and low throughput. Although numerous in vitro tests have been developed, none have successfully reconciled the need for high experimental throughput with the accurate prediction of irritation potential, attributable to the complexity of irritation mechanisms. Simple cell models, while suitable for high-throughput screening, offer limited mechanistic insights, contrasting with more physiologically relevant but less scalable complex organotypic corneal tissue constructs. This study presents a novel strategy to enhance the predictive accuracy of screening-compatible simple cell models in eye irritation testing. Our method combines the results of two in vitro assays—cell apoptosis and nociceptor (*TRPV*1) activation—using micropatterned chips to partition human corneal epithelial cells into numerous discrete small populations. Following exposure to test compounds, we measure apoptosis and nociceptor activation responses. The large datasets collected from the cell micropatterns facilitate binarization and statistical fitting to calculate a mathematical probability, which assesses the compound’s potential to cause eye irritation. This method potentially enables the amalgamation of multiple mechanistic readouts into a singular index, providing a more accurate and reliable prediction of eye irritation potential in a format amenable to high-throughput screening.

## 1. Introduction

The UN Globally Harmonized System (GHS) defines eye damage and irritation as reversible changes to the eye’s anterior surface within 21 days of chemical exposure [1]. Traditional reliance on live animals using the Rabbit Draize eye Test (OECD TG405) is not sustainable due to ethical concerns and legislature changes [2,3]. While ex vivo tests, such as the Bovine Corneal Opacity Test (BCOP) and Chicken Eye Test, preserve the use of whole eye tissues, they have led to over- or under-prediction [2], and are not very amenable for compound screening due to elaborate set-ups required to manipulate the whole organ. Based on current knowledge of the ocular anatomy and the mechanisms underpinning chemical-induced eye irritation, several different in vitro tests have been developed [2,4] and are currently being validated as a substitute for testing with animal ocular samples.

Typically, an in vitro test aims to simulate a specific biological mechanism by which a compound may induce eye irritation. Such tests are comprised of two primary components: (1) the experimental model, which delineates the cellular or tissue framework with which the test compounds interact, and (2) the test read-out, which quantitatively assesses the model’s response to these compounds, examples of which include cell viability, cytokine secretion, or fluorescence dye leakage [5]. The classification of a compound as an eye irritant relies on predefined cut-off values derived from these quantitative read-outs. It has been acknowledged that no single in vitro assay can accurately predict the eye irritation potential of a chemical due to the multiple mechanisms of action. This has spurred the development of complex 3D organotypic corneal tissue models, such as the Epiocular™ assay (OECD TG492) and the Vitrigel-Eye Irritancy test (OECD TG494) [4,6,7]. These models strive to more accurately mimic the physiological structure and functionality of the ocular surface, with the expectation that they will offer a more predictive response to test compounds. Nonetheless, enhancing the physiological relevance of these models introduces greater complexity, costs, and time requirements for conducting in vitro tests [8]. This results in a trade-off regarding experimental throughput and compatibility with automated high-content imaging systems. Simpler models, such as the corneal epithelial cell monolayer utilized in the Short-Term Exposure test (OECD TG491) [9] or the polymer matrix used in the in vitro Macromolecular test (OECD TG496) [10] are more conducive to high-throughput screening but are limited to assessing a single mechanism which cannot predict eye irritation reliably.

We posit that the potential to enhance the accuracy of simple cellular models, such as the corneal epithelial monolayer, lies in the integration of multiple assay readouts addressing various mechanisms of eye irritation. This approach aligns with the prevailing opinion in the field that a diverse array of in vitro tests will be necessary to replace animal-based ocular surface testing [11,12]. However, combining multiple in vitro tests into a cohesive model for predicting eye irritation potential presents a significant challenge, particularly in merging analog read-outs that span various physical measurements, such as cytotoxicity, optical density, and permeability. This challenge can be circumvented with statistical regression analyses, such as logistical regression models and Kaplan–Meier “survival curves”, to facilitate probability-based prediction of event outcomes. The probabilities of several assay readouts can be combined mathematically. However, these regression models necessitate a substantial dataset, typically requiring more than ten observations for the least probable outcome [13,14]. Therefore, statistical regression has not been commonly utilized for analyzing in vitro eye irritation screening assay results, as these are typically performed on bulk cell cultures in microtiter plates or 3D-engineered tissue constructs, which offer limited throughput. To address this limitation, our study employs a cell micropatterning technique to divide a bulk human corneal epithelial cell (hCEC) culture into numerous distinct populations. This enables the measurement of responses to test compounds post-exposure. We selected cell apoptosis and the activation of the nociceptor (pain receptor) Transient Receptor Potential Vanilloid type 1 (*TRPV*1) as indicators of two separate eye irritation mechanisms [15,16]. By miniaturizing and segregating the cell populations, we were able to generate a high volume of response readouts for both cell apoptosis and nociceptor activation. These outcomes are amenable to digitization (i.e., response = 1 or 0), facilitating the assignment of binary outcomes even for small cell populations (i.e., fewer than ten cells). The digitized response data can then be applied to a logistic regression model, enabling the estimation of a concentration-dependent response probability for various test compounds.

## 2. Methods and Materials

### 2.1. Cell Culture

Commercially available primary human corneal epithelial cells (hCECs) (ATCC, Manassas, VA, USA, PCS-700-010, Lot #1170, 5170 and #804003 were used in this study) were cultured in complete primary cell medium prepared by combining corneal epithelial cell growth kit (ATCC, Manassas, VA, USA, PCS-700-040) and corneal epithelial basal medium (ATCC, VA, USA, PCS-700-030) according to manufacturer’s protocol. Cells were sub-cultured after reaching ~80–90% confluence using Trypsin/EDTA solution (ATCC, Manassas, VA, USA, PCS-999-003), and washed twice with 1× Hanks’ balanced salt solution (HBSS, Hyclone, UT, USA, SH30588.01). The cells were subsequently neutralized by Trypsin neutralizing solution (TNS, ATCC, Manassas, VA, USA, PCS-999-004) and re-suspended in the complete medium after centrifugation for 5 min at 1000 rpm and later counted for experimental seeding. Excess cells were cryopreserved at a cell density of 1 million cells/mL with 80% complete medium, 10% Fetal bovine serum (FBS, Hyclone, UT, USA, SV30160.03) and 10% Dimethyl Sulfoxide (DMSO, Life Science, UK, D2660).

Immortalized hCECs (HCE-2 [50.B1], ATCC, Manassas, VA, USA, CRL-11135) were cultured in Keratinocyte-Serum free media containing 0.05 mg/mL bovine pituitary extract (BPE) and 5 ng/mL epidermal growth factor (EGF) (17005042, Gibco, Agawam, MA, USA) and supplemented with 500 ng/mL hydrocortisone and 0.005 mg/mL insulin. The cell culture surface was coated with a solution prepared using HBSS, Fibronectin (0.01 mg/mL, F1141, Sigma Aldrich, Burlington, MO, USA), Collagen (0.03 mg/mL, A1064401, Gibco, Agawam, MA, USA) and Bovine Serum Albumin (0.01 mg/mL, A3311, Sigma Aldrich, Burlington, MO, USA). Cells were sub-cultured after reaching ~80–90% confluence using TrypLE (12605010, Gibco, Agawam, MA, USA), and washed twice with 1× Hanks’ balanced salt solution (HBSS, Hyclone, Logan, UT, USA, SH30588.01). The cells were subsequently neutralized with Low Glucose DMEM with 10% FBS and re-suspended in the complete medium after centrifugation for 5 min at 1000 rpm and later counted for experimental seeding. Excess cells were cryopreserved at a cell density of 1.5 million cells/mL with 50% complete medium, 40% FBS and 10% DMSO.

HEK293-*TRPV*1 cell line with Red Fluorescence Protein (RFP) and Puromycin dual marker (SC027-RB, GenTarget Inc., San Diego, CA, USA) were cultured in High Glucose DMEM (11965084, Gibco, Agawam, MA, USA) supplemented with non-essential amino acids (1:100) and blasticidine 1:1000 (15205-25MG). Cells were sub-cultured after reaching ~80–90% confluence using TrypLE (12605010, Gibco, Agawam, MA, USA), post twice washing with 1× Hanks’ balanced salt solution (HBSS, Hyclone, UT, USA, SH30588.01, USA). The cells were subsequently neutralized with Low Glucose DMEM with 10% FBS and re-suspended in a complete medium after centrifugation and later counted for experimental seeding. Excess cells were cryopreserved at a cell density of 2.25 million cells/mL with 50% complete medium, 40% FBS and 10% DMSO.

All cells were maintained at 37 °C in a humidified 5% carbon dioxide (CO_2_) environment with medium replacement on alternative days.

### 2.2. Cell Micropatterning

hCECs were seeded onto commercial micropatterned glass substrates (CYTOO, Grenoble, France) which have been customized to contain an array of circular micropatterns (65 and 80 µm diameter) on which cells can adhere. The area surrounding the circular micropatterns was selectively passivated to attenuate cell and protein adhesion. Before cell seeding, the micropatterned substrates were coated with a mixture consisting of Fibronectin (Sigma-Aldrich, Burlington, MO, USA F1141 0.03 mg/mL), Bovine Collagen I (5 mg/mL Gibco, Agawam, MA, USA, A1064401, 0.01 mg/mL) and Bovine Serum Albumin (BSA, Sigma-Aldrich, St. Louis, MO, USA, A331, USA, 0.01 mg/mL) in 1× HBSS, and incubated at 37 °C for 24 h. Coated micropattern substrates were transferred to a 35 mm dish (Thermo Scientific, China, 150460), which was passivated with 10% Pluronic acid (Sigma-Aldrich, St. Louis, MO, USA, P2443) in 1× HBSS overnight. Cells were seeded onto the micropatterned substrates at a density of 300,000 cells/mL. Unattached cells were removed after 2 h by washing with 1× HBSS 5 times, followed by passivation with 0.2% Pluronic acid for 10 min, and finally replaced with 2 mL of complete medium. The cells were cultured on the micropattern substrates for 24–48 h based on the assay being performed.

### 2.3. Test Compound Treatment for Micropatterned Cells

Cells seeded on the micropatterned glass substrates were assembled into a specialized CYTOO chamber (CYTOO, Cambridge, MA, USA, 30-011) to partition a single micropatterned glass substrate into 4 wells. This allowed for 4 different compound treatments on the same micropatterned substrate. The chamber consisted of a bottom plate to fit the micropatterned substrate. The bottom plate was then magnetically assembled into the main body, which was locked using a silicon gasket. The silicon gasket and the main body partitioned the micropatterned glass substrate (i.e., CYTOO chip) into 4 wells. A glass lid was placed at the top to cover the whole setup and keep it sterile (Appendix A). For each compound treatment experiment, every CYTOO chip was administered with 3 different compound concentrations and 1 vehicle control as an internal control to minimize inter-experimental variation.

### 2.4. Bulk Apoptosis Assay

Primary hCECs were plated in 96 well plates at a density of 20,000 cells/well. After 24 h of culture, cells were treated with 100 µL of each of the concentrations of staurosporine (positive apoptosis inducer), dissolved in DMSO to obtain the test solutions of 0.1–10^4^ nM with serial dilutions prepared in a complete medium. As a vehicle control, 0.1% DMSO in culture medium was used. Sodium lauryl sulphate (SLS), an OECD category 1 compound, was dissolved 1× HBSS to obtain test solutions of 10^3^–10^8^ nM. As a vehicle control, HBSS solution in the culture medium was used. The untreated category of cells was exposed to only media. Post drug treatment, the cells were labelled using a staining solution consisting of Hoechst 33342 (cell nucleus stain, Thermo Fisher–Scientific, Waltham, MA, USA, 62249, 1:500), CellEvent^TM^ Caspase-3/7 green detection reagent (Thermo Fisher–Scientific, Waltham, MA, USA, C10423, 1:1000) and Ethidium homo dimer (necrotic cell stain, Sigma-Aldrich, St. Louis, MO, USA, cat. 46043, 1:500) dissolved in 1× HBSS. Cells were incubated with the staining solution for 40 min at 37 °C. The cells were subsequently fixed post-washing with 1× HBSS containing 4% Paraformaldehyde (PFA) for 10 min at room temperature. The wells were subsequently imaged using a Nikon Fluorescence microscope. The percentage of apoptotic cells (caspase stained) and dead cells (Ethidium homo dimer stained) were calculated.

### 2.5. Apoptosis Assay on Micropattern Substrates

The apoptosis assay was performed 24 h post-seeding. Before drug treatment, the micropatterned substrates were removed from 35 mm dish before being assembled and partitioned into 4 replicate wells using CYTOO chambers as described in Section 2.3. The micropatterned hCECs were treated with staurosporine (positive apoptosis inducer) and dissolved in DMSO to obtain the test solutions of 0.1–10^6^ nM with serial dilutions. As the vehicle control, 0.1% DMSO in the culture medium was used. Sodium lauryl sulphate (SLS), OECD category 1 compound, dissolved in 1× HBSS used to obtain test solutions of 100–10^7^ nM. As the vehicle control, HBSS solution in the culture medium was used. Ethyl methyl acetoacetate (EMA), OECD Category 2B compound dissolved in DMSO to obtain the test solutions of 500–5 × 10^6^ nM with serial dilutions. As the vehicle control, 0.1% DMSO in the culture medium was used. Drug treatment was conducted for 24 h at 37 °C. A vehicle control was included in one of the four CYTOO chambers for each micropattern substrate.

After treatment with test compounds, the cells were stained using a staining solution consisting of Hoechst 33342 solution (cell nucleus stain, Thermo Fisher–Scientific, Waltham, MA, USA, 62249, 1:500), cell event caspase-3/7 green detection reagent (Thermo Fisher–Scientific, Waltham, MA, USA, C10423, 1:1000) and Ethidium homo dimer (dead cell stain, Sigma-Aldrich, St. Louis, MO, USA, cat. 46043, 1:500) dissolved in 10% FBS with 1× HBSS. Then, cells were incubated with the staining solution for 40 min at 37 °C. The cells were subsequently fixed post washing with 1× HBSS with 4% Paraformaldehyde (PFA) for 10 min at room temperature. The culture chambers were removed and disassembled to retrieve the micropatterned substrates, which were then mounted on glass coverslips for imaging. The cells were imaged at 10× magnification using a Nikon Eclipse Ti Fluorescence microscope, NY, USA.

### 2.6. Quantification and Digitization of Apoptosis Assay Readout

Images from the 2 fluorescent channels (green: Caspase3/7 and blue: DAPI) were quantified and batch analyzed using an ImageJ (Version 1.51) macro [17,18]. The macro first loads a predefined set of regions of interest (ROIs) that serve as initial locations of the micropatterns. To account for sight variation in the plate placement, iterations of random small displacement were applied to this set of ROIs to optimize for the maximum area of nuclei. Next, each ROI was used to count the nuclei per micropattern from the DAPI channel image that had binarized by auto-threshold. The SetOption (“BlackBackground”, true) function from Image J was used to remove the background signal in the image before the auto-threshold and binarization functions were performed. Next, each ROI was enlarged by 2 pixels to measure the fluorescence intensity of the green caspase3/7 channel. Eventually, the fluorescence intensity of the caspase staining (apoptosis indicator) was determined for individual cells within every micropattern. The caspase intensity was obtained for vehicle control as well as each of the drug treatment groups. The caspase intensity of every compound-treated sample was subtracted from the average caspase intensity of its corresponding vehicle control. The normalized fluorescence intensity was used for data binarization, wherein intensities above 20% of the averaged vehicle control were defined as positive apoptosis (“1”) and values below 20% of the averaged vehicle control were defined as negative apoptosis (“0”). Each micropattern was defined as apoptotic (“1”) if ≥50% of cells in the micropattern were apoptotic (“1”) and vice versa for non-apoptotic micropatterns.

### 2.7. Calcium Influx TRPV1 Channel Activation Assay

After 24 h of seeding on CYTOO chips, primary hCECs were labelled with Fluo-4 AM (F14201, Invitrogen, OR, USA, 2 µM concentration) and Hoechst (1:500) in phenol red-free DMEM (11054001, Gibco, Agawam, MA, USA) supplemented with 10% FBS and incubated at 37 °C for 45 min. Labelled cells were then recovered in phenol red-free DMEM for 30–40 min at 37 °C before replacing with fresh medium for imaging. Live-cell imaging was performed using a laser scanning confocal microscope (LSM800 Airy Scan, Carl Zeiss, Jena, Germany) at 10× magnification for a period of 5 min at 5 frames per second. Test compounds at designated concentrations were administered approximately after 1 min to obtain the baseline signals. The cells were treated with a positive *TRPV*1 activator, capsaicin (10 µM), a category 1 compound SLS (100–10^6^ nM) and a category 2B compound EMA (500–5 × 10^6^ nM) with their respective vehicle controls, i.e., DMSO, HBSS and DMSO, respectively. A vehicle control was included in one of the four CYTOO chambers for each micropattern substrate.

### 2.8. Quantification and Digitization of TRPV1 Activation Assay Readout

Images from the two fluorescent channels (green: Fluo-4 and blue: DAPI) were quantified and analyzed using an ImageJ macro similar to the apoptosis assay. The micropatterns were identified using the pre-defined ROI and cell nuclei per micropattern were determined using the “binarization” function on the DAPI channel image. Each nuclei mask was enlarged by 2 pixels to obtain the mask for the Fluo-4 image. The Fluo-4 fluorescence intensities of individual cells in every micropattern were quantified at each time frame. The Fluo-4 intensity was obtained for vehicle control as well as each of the drug treatment groups.

The Fluo-4 fluorescence intensity of each cell was plotted against the time frames to obtain the calcium influx response of each cell. Since the baseline Fluo-4 intensity of each cell is highly heterogeneous due to variability in the intracellular Ca^2+^ concentration, we normalized the intensity of each image frame by subtracting the baseline intensity at the initial frame (at $time=\;t_{0}$). The change in intracellular Ca^2+^ concentration induced by a test compound was determined by calculating the integrated peak area between the time of induction ($t_{1f,\;ind}$) and 20 frames after induction ($t_{20f,\;ind}$) (~4 s post induction) as follows:$$Integrated\;Peak\;area=\int_{t_{1find}}^{t_{20find}}Normalized\;Intensity\;dt$$

A positive integrated peak area was digitized to “1” binary outcome, while a negative integrated peak area was binarized to “0”. Each micropattern was defined as a positive *TRPV*1 activator (“1”) if ≥50% cells in the micropattern had a positive outcome and vice versa for non-activator of *TRPV*1.

### 2.9. Statistical Analysis and Logistical Regression Modelling

Each experiment was carried out in 3 biological repeats unless otherwise mentioned. The digitized responses (i.e., cell apoptosis or Ca^2+^ influx) were fitted into a Logistic Regression Model (Prism version 9, GraphPad, Boston, MA, USA) to determine the response probability as a function of test compound concentration. The collective probability of a test compound causing eye irritation $\left(P_{eye\;irritation}\right)$ by either inducing cell apoptosis ($P_{apoptosis}$) or activating the nociceptor ($P_{TRPV1}$) at a given concentration was determined using the following formula, assuming that the events are non-mutually exclusive (i.e., apoptosis and *TRPV*1 action can occur at the same time):$$P\;\left(eye\;irritation\right)=P\left(apoptosis\right)+P\left(TRPV1\right)-P(apoptosis\;\cap \;TRPV1)$$

## 3. Results and Discussion

### 3.1. Partitioning of Human Corneal Epithelial Cells Using Cell Micropatterning

Micropatterned glass substrates, which are differentially passivated [14,19] were employed to partition the human corneal epithelial cell (hCECs) population into an array of circular micropatterns. We evaluated micropatterns of different sizes to determine their suitability of patterning primary and immortalized hCECs as well as to quantify the number of cells per micropattern. Primary and immortalized hCECs were first generated on 1000 µm circular micropatterns, and the stability of each cell type on micropatterns was monitored over time. We observed that primary hCECs were able to maintain a confluent monolayer over a period of 7 days whereas the immortalized cell line lost contact inhibition and grew as multiple cell layers after 3 days (Figure 1A). The multiple cell layers impose technical challenges to subsequent cell imaging-based assays. Furthermore, the immortalized hCEC micropatterns became unstable once they had multiple cell layers and were more prone to detach after prolonged culture.

Micropatterns of diameters 1000 µm, 500 µm, 225 µm, 140 µm and 80 µm were also evaluated with both primary and immortalized hCECs in order to determine the optimal micropattern size that will give us the desired number of cells per micropattern. A smaller number of cells per micropattern (i.e., <10 cells per micropattern) will allow us to easily determine the 50% response threshold required to binarize the output. We were able to successfully generate a confluent cell monolayer across the range of micropattern sizes for both the cell lines (Figure 1B). It was observed that the 80 µm diameter micropattern yielded colonies of cell which were not over clustered, and thus the number of cells per micropattern could be accurately quantified by labelling the cell nuclei with DAPI (Figure 1C). Both primary and immortalized hCEC micropatterns contained a similar average of 4.5 ± 2 and 4.2 ± 2 cells per 80 µm diameter micropattern, respectively (Figure 1D). However, since primary hCECs were able to maintain a stable confluent monolayer of cells on the micropatterned substrate over a longer period, we deemed that they were more suitable for conducting eye irritation assays.

### 3.2. Digitization of Apoptosis Assay Using Primary hCEC Micropatterns

Once successful partitioning of primary hCECs into a large number of discrete 80 µm circular micropatterns was achieved, we could proceed to digitize the caspase 3/7 apoptosis assay and fit the binarized response data after compound treatment into a binary logistical regression model. Positive caspase 3/7 and DAPI staining in each cell within a micropatterned colony could be clearly visualized and demarcated after treatment with a known apoptosis inducer—staurosporine (STS) (Figure 2A). A predefined micropattern array mask was overlayed on each image to identify each micropattern area. This enabled automated enumeration of live and dead cells inside each micropattern using the nuclei stain and caspase 3/7 intensity per cell (Figure 2B).

To digitize the assay readout, the apoptotic intensity per cell for each treatment condition was normalized to the vehicle control to set the threshold for binarizing of response outcome of a single micropattern. A micropattern has been defined as apoptotic (binary score: 1) if >50% of cells had caspase 3/7 intensity more than 20% of the vehicle control and vice versa for a non-apoptotic micropattern (binary score: 0). The binarized apoptosis responses of every micropattern after treatment with the range of compound concentrations were fitted into a logistic regression model to determine the probability that a particular compound will induce apoptosis in primary hCECs after 24 h.

We then employed the digitized apoptosis assay to evaluate an OECD Category 1 compound, sodium lauryl sulphate (SLS) and Category 2 compound, ethyl methyl acetoacetate (EMA) across a concentration range of 100–10^6^ nM, where dose responses were observed in a population-averaged bulk analysis (Figure 2C). For SLS, the conducted binary logistic regression yielded the following probability equation: $P\left(1\right)=\frac{\exp\left(-1.1057+0.000001C\right)}{1+\;\exp\left(-1.1057+0.000001C\right)}$ (Figure 2D), where *P* = probability and *C* = concentration. The regression curve of SLS agreed with the bulk assay for the analogue readout of % apoptotic cells when treated with of 1000–10^8^ nM SLS (Figure 2C). Both the curves followed a similar trend, indicating that logistical regression of binarized apoptosis response is a promising approach that corresponded well to conventional bulk cytotoxicity assays with analogue readouts. Based on the logistic fitting of the dose response curve for SLS, we could determine that the IC_50_ concentration corresponding to a 0.5 response probability was ~1 × 10^6^ nM (1 mM) (Figure 2D). When compared to the short-term exposure (STE) assay, which typically evaluates compounds at two concentrations, namely 0.05% and 5% *w*/*w* [20], our IC_50_ value was much lower than the lower concentration limit used in STE (0.05% *w*/*w* for SLS = 1.73 × 10^7^ nM). Previous studies have shown that the percentage cell viability of SLS relative to untreated controls was <1% at both 0.05 and 5% *w*/*w* concentration [21]. At 0.05% *w*/*w* (1.73 × 10^7^ nM), the predicted apoptosis probability is >0.99 (Figure 2D), which is in strong agreement with the former report. This confirmed that SLS is a potent cytotoxic agent, and likely induces permanent eye damage via this mechanism.

For EMA, the binary logistic regression yielded the following probability equation: $P\left(1\right)=\frac{\exp\left(-0.9490+0.0C\right)}{1+\;\exp\left(-0.9490+0.0C\right)}$, which was independent of EMA concentration (Figure 2E). Our data suggested that EMA was mildly cytotoxic towards primary hCECs. This was in agreement with reported STE studies, whereby the cytotoxicity of EMA was only apparent at high concentrations of 5% (3.5 × 10^9^ nM) [22], which is beyond the range of concentrations evaluated in this study. At 0.05% (3.5 × 10^7^ nM), the STE assay showed that cell viability remained at almost 100% [22], which concorded with our observations. Although EMA (categorized as compound 2B in GHS) was classified as a moderate toxicant based on the STE assay, this can be attributed to the high concentration being considered (5%), and cytotoxicity may not be the dominant mechanism through which EMA causes eye irritation.

### 3.3. Establishment of Nociceptor Activation Assay in Primary hCECs

Nociception is another mode of action for eye irritation, where noxious eye irritants can stimulate periphery neurons manifested as pain [23]. Transient receptor potential vanilloid 1 (*TRPV*1) receptor activation was previously used as an indicator of nociception in *TRPV*1 overexpressing neuroblastoma cells treated with different irritant compounds [23], although this has not been demonstrated in hCECs. Thus, we first characterized *TRPV*1 expression in primary hCECs (Figure 3B) and benchmarked to that of a *TRPV*1 overexpressing HEK293 cell line (*TRPV*1-HEK293) (Figure 3A). Fluorescence imaging showed that *TRPV*1 was expressed in both cell types tested (Figure 3A,B). Since primary hCECs expressed appreciable levels of *TRPV*1 and could form stable micropatterns, we could proceed to establish *TRPV*1 activation assay using the micropatterned primary hCEC cultures.

Calcium influx is commonly used as an indicator of *TRPV*1 activation [24], which is measured using changes in the intensity of Fluo-4, a calcium-sensitive dye. Capsaicin was employed as a positive *TRPV*1 activator. Cultured primary hCECs stained with Fluo-4 were induced with 10 µM capsaicin and the changes in the fluorescent signals were recorded over time. A sharp rise in fluorescent intensity of Fluo-4 was detected at the time of capsaicin induction, measured as peak amplitude (Figure 3C). With different concentrations of capsaicin, a dose-dependent response of Ca^2+^ influx in primary hCEC was observed, whereby peak amplitudes of Fluo-4 fluorescence intensity increased with higher capsaicin concentration (Figure 3D). Hence, it can be shown that *TRPV*1 activation in primary hCECs can be measured by intracellular calcium fluxes upon induction with chemical activators.

### 3.4. Establishment of Digitized TRPV1 Activation Assay in Primary hCECs

Digitization of the *TRPV*1 activation assay was conducted using the same test compounds as for the cytotoxicity assessment. Calcium influx was measured to evaluate the dose–response of micropatterned primary hCECs 24 h post seeding, upon compound induction after 1 min of baseline imaging. The hCECs were induced with SLS (100–10^6^ nM) and EMA (500–5 × 10^6^ nM) with their vehicle controls, i.e., HBSS and DMSO, respectively. We observed that hCECs displayed significant increases in intracellular Ca^2+^ fluxes in response to both SLS and EMA. However, contrary to capsaicin, which generated a sharp peak of calcium influx following compound exposure, broad calcium peaks were observed for the tested eye irritants (Appendix A). This prevented simple measurements of peak amplitudes to indicate Ca^2+^ influx. Moreover, individual cells had a varying baseline of the intracellular Ca^2+^ levels (Appendix A). To circumvent these challenges, baseline-normalized integrated peak area within a pre-defined time period (i.e., 20 frames after induction, ~4s post induction) was quantified to determine the change in intracellular Ca^2+^ that was induced by the addition of the test compounds (Appendix A). The resultant integrated peak area was used to digitize the *TRPV*1 activation response of individual cells within a micropattern. A positive integrated peak intensity was assigned a value of “1” while a negative integrated peak intensity was assigned a value of “0” (Appendix A). If >50% of cells in a micropattern have been assigned “1”, the micropattern was considered as a positive activator of *TRPV*1 and vice versa for a non-activator micropattern. These binary values were then fitted into a logistic regression plot to determine the response probability of different eye irritants.

The probability of *TRPV*1 activation induced by SLS, as determined by logistic regression fitting, was ~0.5 across the range of concentrations investigated (Figure 4A). In the case of EMA, the probability increased to 0.8 across all concentrations tested, with a slight decrease at the highest concentration tested (5 × 10^7^ nM) (Figure 4B). However, both the compounds failed to elicit a dose-dependent Ca^2+^ influx response in primary hCECs. To examine if this apparent lack of dose dependency in the probability *TRPV*1 activation assay could be due to the large variation observed in the Ca^2+^ influx responses of single cells [25], we examined the integrated peak area/cell of Fluo-4 intensity for both the compounds. At a 1 × 10^5^ nM concentration of SLS, the magnitude of calcium influx was significantly higher than at lower concentrations, with reduction in magnitudes at 1 × 10^6^ and 1 × 10^7^ nM, which can probably be attributed to a decrease in cell viability in the hCECs (earlier apoptosis results) (Figure 4C). For EMA, an increase in the intracellular calcium levels was also observed, with the highest response at 5 × 10^6^ nM; however, similar to SLS, this was not at the highest concentration tested (Figure 4D).

This apparent lack of dose dependency is in contrast with Lilja et al., [23] whereby a dose-dependent effect of SLS on *TRPV*1 was observed. This could be due to different endpoints being measured as well as the use of immortalized neuroblastoma cell lines, which could be less sensitive and susceptible to the cytotoxic effects of test compounds. Nevertheless, our data are in line with Lehmann, Hatt and van Thriel [26] where the dose–response for tested compounds was also not observed using the calcium imaging method but was only apparent using a voltage clamp method. The authors concluded that calcium imaging may be suitable as an initial screening approach for irritant potency assessment, but dose responses may need to be confirmed by another complementary assay.

As there is a lack of data in the current literature that reports the effect of EMA across multiple concentrations, we were unable to verify our observations on the concentration effect of EMA on nociception. Nonetheless, EMA has been successfully identified as an eye irritant by different more complex in vitro tests [27]. Therefore, while the cytotoxic effect of EMA was not pronounced, it appears that nociceptor activation may be a more dominant mechanism of action for causing eye irritation.

### 3.5. Combined Probability of Digitized Readouts from Apoptosis and TRPV1 Activation Assay

To determine the eye irritant category of each of the compounds, the probability from the two assays was combined to estimate an integrated probability of eye irritation mediated through cytotoxicity and/or nociceptor activation across different concentrations. The combined probability was determined assuming that the events are non-mutually exclusive (i.e., apoptosis and TRPV1 action can occur at the same time).

Our results revealed that below the cytotoxic concentration of SLS (<10^6^ nM), EMA appeared to have a higher likelihood (0.85) of causing eye irritation, manifested as eye stings through nociceptor activation (Figure 5). In comparison, eye irritation probability of SLS below 10^6^ nM was only ~0.6. Our results are consistent with the observation that the eye irritation potential of SLS appeared to be highly dependent on the concentration used for hCEC induction [28].

We found that the combined digitized apoptosis and nociception readouts were able to differentiate SLS and EMA quantitatively. This contrasts with existing in vitro assays, such as the STE, which qualitatively assigns compounds into categories [9]. Results showed that SLS is a potent cytotoxic agent and thereby capable of inducing non-reversible damage, typical of a GHS category 1 compound. Hence, our method combines the advantageous features of STE (i.e., simplicity and high throughput) with the predictive potential of more complex organotypic eye models to differentiate category 1 and 2 eye irritants. The probability-based eye irritation testing approach presented here is closely aligned with the principles outlined in the newly established OECD Test Guideline 467, “Defined Approaches for Serious Eye Damage and Eye Irritation” [29,30]. This guideline underscores the challenge of identifying eye irritation potential due to the often unknown modes of action for many chemicals, which renders single tests insufficient. To address this, the guideline advocates for the integration of multiple in vitro methods through a Defined Approach (DA), utilizing fixed data interpretation procedures (such as mathematical models or rule-based approaches) across diverse data types (including in silico predictions, in chemico, and in vitro data) from specific information sources. Our method, by being able to account for distinct mechanisms of eye irritation, significantly enhances its ability to detect potential irritants among new compounds. Future work to augment our platform’s predictive accuracy can do so by incorporating digital data representing other biological processes like cell adhesion and inflammation, further aligning our approach with OECD TG467’s comprehensive strategy for assessing eye irritation potential.

## 4. Conclusions

In conclusion, this study introduces a novel method for assessing eye irritation risk by merging probability-based responses from two in vitro tests—cytotoxicity and nociception activation. Utilizing micropatterned primary human corneal epithelial cells (hCECs), we digitized assay outcomes for *TRPV*1 pain receptor activation and apoptosis. These digitized results were then integrated using logistic regression to produce combined response probabilities. This innovative approach successfully differentiated the eye irritation potential of various compounds, exemplified by Sodium Lauryl Sulfate (SLS, category 1) and Ethanolamine (EMA, category 2B), demonstrating its potential as a reliable and advanced tool for chemical safety assessment.

## Figures and Tables

**Figure 1 bioengineering-11-00315-f001:**
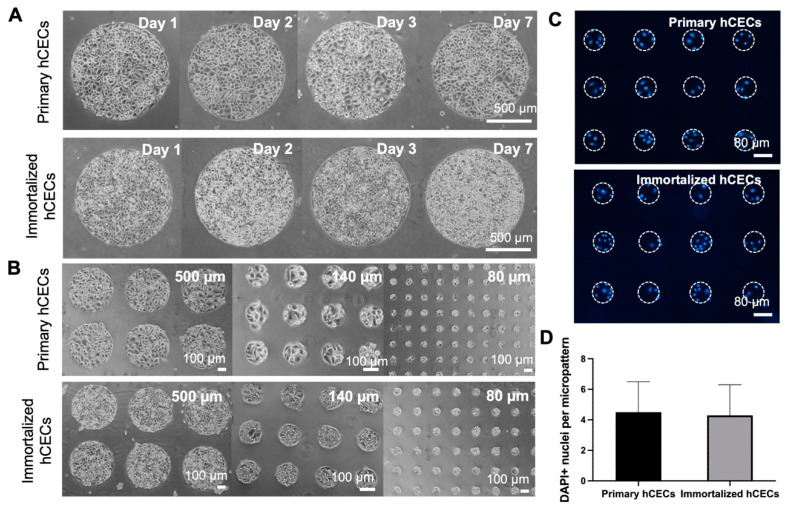
**Cell micropatterning for partitioning human corneal epithelial cell (hCEC) cultures.** (**A**) Phase contrast images showing the culture of primary and immortalized hCECs on 1 mm diameter circular micropatterned glass substrates over time. (**B**) Primary and immortalized hCECs on micropatterns of different diameters at 24 h post patterning. (**C**) Nuclear staining (DAPI^+^) of hCECs on 80 µm micropatterns to quantify number of cells per micropattern. (**D**) Quantification results of average number of cells per 80 µm micropattern. Data are average ± s.d. of >1000 micropatterns.

**Figure 2 bioengineering-11-00315-f002:**
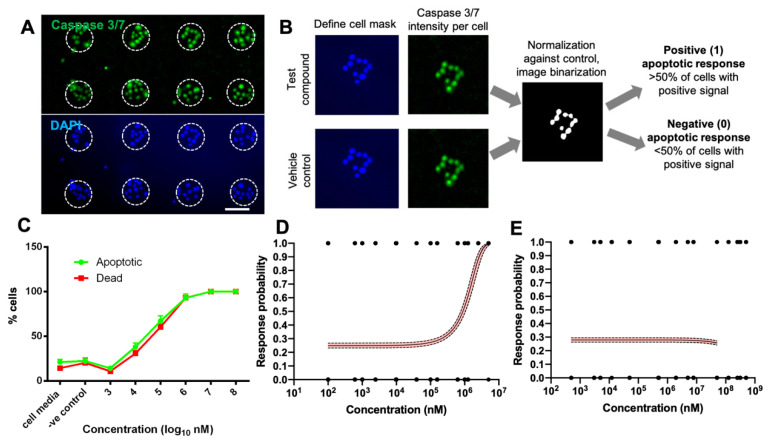
**Establishment of probability-based apoptosis assay.** (**A**) Caspase 3/7 (top panel) and nuclei (bottom) staining on micropatterned primary hCECs after 24 h of treatment with Stauorosporin (STS), a paradigm apoptosis inducer. Dotted circles denote individual micropatterned cell populations from which binary (1/0) apoptotic responses were recorded for logistical fitting. (**B**) Process flow for digitization of apoptosis response of individual micropatterned cell populations. (**C**) Bulk assay with an analogue readout depicting % of cells which were positive for Caspase 3/7 (apoptotic cells) and EthD (dead cells) when primary hCECs were treated with SLS at varying concentrations. Data are average ± average SD of three independent experiments. (**D**,**E**) Digitized assay with probability readouts by fitting the binarized responses of individual micropatterns into a logistical regression model. Dotted lines represent 95% confidence interval of the regression model. (**D**) Primary hCECs after 24 h treatment with SLS. (**E**) hCECs after 24 h treatment with EMA. Data in (**D**,**E**) are fitted from >200 micropatterns obtained from at least two independent experiments per concentration.

**Figure 3 bioengineering-11-00315-f003:**
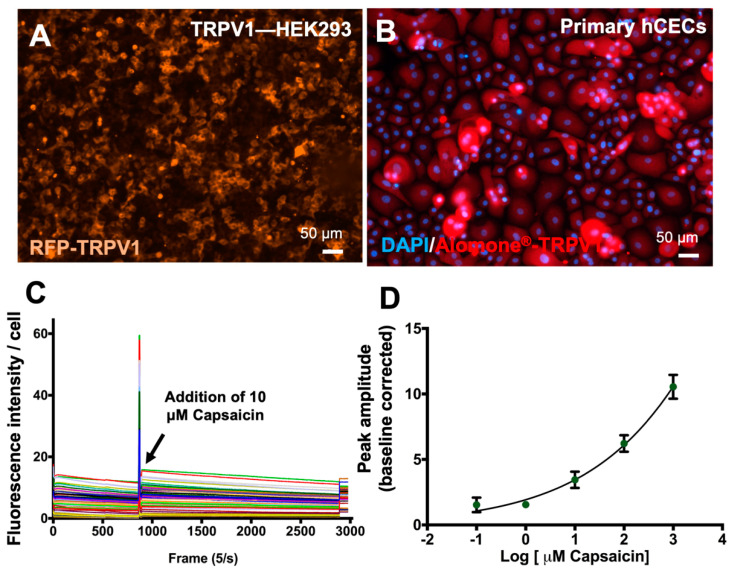
**Establishment of *TRPV*I nociceptor activation assay.** (**A**,**B**) Immunofluorescence images showing expression of the nociceptor TRPV1 in different cell types: (**A**) HEK293 cells overexpressing *TRPV*1 and (**B**) primary hCECs. (**C**) Change in Fluo-4 fluorescence intensity per cell over time indicated intracellular Ca^2+^ influx triggered by 10 μM Capsaicin, a paradigm TRPV1 activator in primary hCECs. (**D**) Change of Fluo-4 fluorescence intensity peak amplitudes with different Capsaicin concentrations showing Ca^2+^ influx due to *TRPV*1 activation in primary hCECs.

**Figure 4 bioengineering-11-00315-f004:**
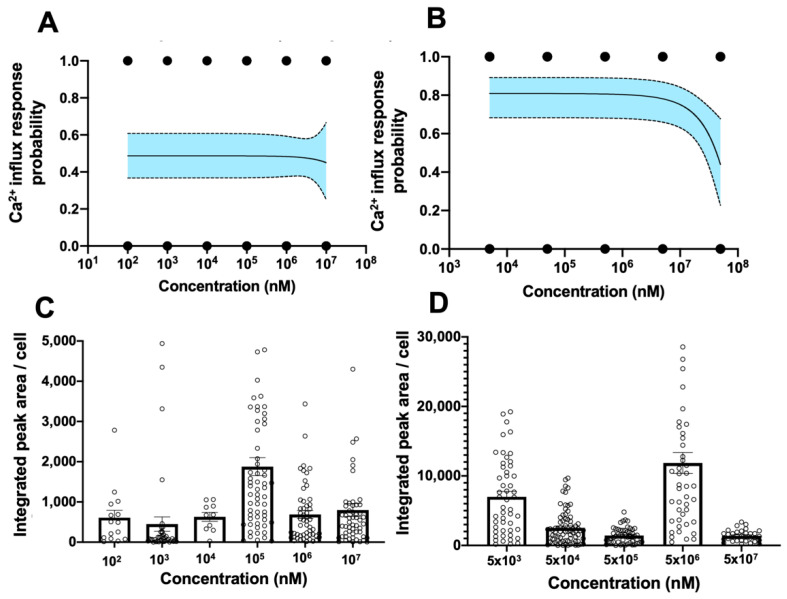
**Probabilities of Ca^2+^ influx in micropatterned primary hCECs in response to different eye irritants.** (**A**,**B**) Logistical regression of binarized Ca^2+^ influx response in individual hCEC micropatterns after treatment with (**A**) SLS and (**B**) EMA at varying concentrations. Dotted lines represent 95% confidence interval of the regression model. Data are fitted from at least 10 micropatterns obtained from two independent experiments per concentration. (**C**,**D**) Integrated peak area of Fluo-4 intensity per cell showing strength of Ca^2+^ response after treatment with (**C**) SLS and (**D**) EMA at varying concentrations. Data in (**C**,**D**) are average +/− sem of 10–100 single cells collected from two experiments.

**Figure 5 bioengineering-11-00315-f005:**
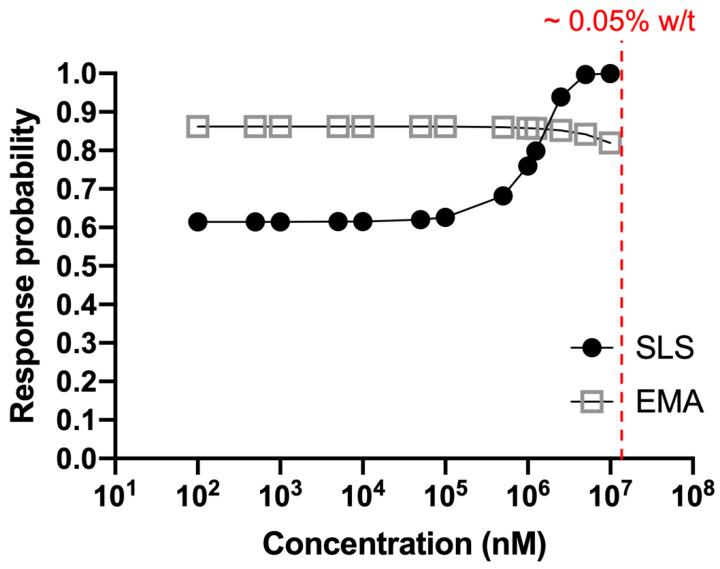
Combined probability of apoptosis and calcium influx assays for different concentrations of SLS and EMA.

## Data Availability

The data that support the findings of this study are available from the corresponding author, YCT, upon reasonable request.

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
