# Peer review of "Development of a Probability-Based In Vitro Eye Irritation Screening Platform"

_bioengineering, 2024, doi:10.3390/bioengineering11040315_

Round 1

Reviewer 1 Report

Comments and Suggestions for Authors

To determine the eye irritation capability of different test compounds, this study relies on the combined outcomes of two different in vitro assays cell apoptosis and activation of the nociceptor, Transient Receptor Potential Vanilloid type 1 (TRPV 1). It is very promising screening model for eye irritants but the suggested in vitro model has to be improved and a number of chemicals from each irritant class has to be tested for validation. 

R1. Page 3, Line 125-127. What is the main expected advantages of seeding hCECs onto commercial micropatterned glass substrates comparing with 96-well plate?. It seems that the use of CYTOO chip including its assembles might increase much more contamination risks. 

R2. Page 4, Line 153. What is the effects of using Primary hCECs monolayer versus using reconstructed in vitro Epiocular™ model (OECD TG492) on the percentage of apoptotic cells?. Which one might have more positive false results?

R3. Page 6, Line 271-284. Cell micropatterning (i.e. <10 cells per micropattern) could be accurately quantify by labelling the cell nuclei. However, as you mentioned that maintaining as stable confluent monolayer of cells on the micropatterned substrate over a longer period of time is more suitable for conducting the eye irritation assays.

Please justify this dilemma, why did you use CYTOO chip if is not sufficient enough for conducting assay.

R4. Fluo4 was in some places written as Fluo-4 throughout the manuscript. It needs to be corrected.

R5. Page 9, Line 433-442. This study suggested as a novel method for assessing eye irritation, therefore it has to be compared with the existing OECD TG (405, 467, 494, 496, 492). The advantages, disadvantages and limitations should be discussed.

Comments on the Quality of English Language

The consistency of the words throughout the manuscript needs to be checked. Quality of English is average and needs to be improved.

Reviewer 2 Report

Comments and Suggestions for Authors

The authors developed in vitro eye irritation screening platform. The proposed technique relies on the combined outcomes of two different in vitro assays namely cell apoptosis and activation of the nociceptor, Transient Receptor Potential Vanilloid type 1 (TRPV 1).

The topic of the paper sounds interesting the following comments should be addressed before further processing:

1-       Why authors have used 2D in vitro cell culture models, how about using organoids? Spheroid models which has a superior compatibility with in vivo models

2-       The images ( Figure 2, 3) for the cells seems have some backgrounds, how authors considered it in their calculations, can authors improve the quality of figures for cells?

3-       In figure 4C-D seems there is a high variation, what can be the reason for that?

4-       In the introduction section, can authors explain more about the novelties of this work and discuss what challenges in the field are addressed with this article.

5-       More discussion on results can be added to the manuscript.

6-       How about necrosis analysis? Is it relevant for this work to evaluate the necrosis

7-       It is recommended to discuss the potential organ on a chip to provide advanced in vitro models for this study for future works. ( in discussion section).

Comments on the Quality of English Language

it can be improved

Round 2

Reviewer 2 Report

Comments and Suggestions for Authors

Authors addressed all my comments.